# Multi-Agent Decision S4: Leveraging State Space Models for Offline Multi-Agent Reinforcement Learning

## Abstract

Goal-conditioned sequence-based supervised learning with transformers has shown promise in offline reinforcement learning (RL) for single-agent settings. However, extending these methods to offline multi-agent RL (MARL) remains challenging. Existing transformer-based MARL approaches either train agents independently, neglecting multi-agent system dynamics, or rely on centralized transformer models, which face scalability issues. Moreover, transformers inherently struggle with long-term dependencies and computational efficiency. Building on the recent success of Structured State Space Sequence (S4) models, known for their parameter efficiency, faster inference, and superior handling of long context lengths, we propose a novel application of S4-based models to offline MARL tasks. Our method utilizes S4's efficient convolutional view for offline training and its recurrent dynamics for fast on-policy fine-tuning. To foster scalable cooperation between agents, we sequentially expand the decision-making process, allowing agents to act one after another at each time step. This design promotes bi-directional cooperation, enabling agents to share information via their S4 latent states or memory with minimal communication. Gradients also flow backward through this shared information, linking the current agent's learning to its predecessor. Experiments on challenging MARL benchmarks, including Multi-Robot Warehouse (RWARE) and StarCraft Multi-Agent Challenge (SMAC), demonstrate that our approach achieves competitive or improved performance compared to state-of-the-art offline RL and transformer-based MARL baselines across most tasks.

## 1 Introduction

Multi-agent reinforcement learning (MARL) has demonstrated significant success in learning complex policies that require coordination among multiple agents to maximize a shared objective (Cao et al., 2012; Berner et al., 2019; Ye et al., 2015). However, this success often relies on a substantial number of interactions with the environment, which can be computationally expensive in high-fidelity simulations or prohibitively risky in real-world applications. To enhance sample efficiency, offline reinforcement learning (RL) algorithms (Lee et al., 2021; Fujimoto et al., 2019; Kumar et al., 2019; 2020; Kostrikov et al., 2021; Xu et al., 2022a; Li et al., 2022; Xu et al., 2023) have been developed, enabling learning from pre-collected offline datasets, thus reducing the need for extensive online interactions.

Offline RL is plagued by the well-known issue of distribution shift, which leads to extrapolation errors when encountering out-of-distribution (OOD) samples during policy training. This occurs when the learned policy deviates from the unknown behavior policy used to collect the training data. To mitigate this, various forms of regularization (Kumar et al., 2019) are introduced to ensure that the learned policy remains close to the behavior policy (Kumar et al., 2020; Xu et al., 2023; 2022a). In multi-agent settings, the joint state-action space expands exponentially as the number of agents increases. This makes it challenging to apply these regularization techniques globally on the joint state-action space, leading to sparse and less effective regularization constraints, especially when working with a limited and less diverse offline dataset.

Sequence-based supervised learning has been concurrently applied to address offline MARL, leveraging the significant success of supervised learning in capturing complex patterns from large offline datasets. This

approach, first introduced by the Decision Transformer (DT) (Chen et al., 2021), has demonstrated its ability to learn policies in an autoregressive fashion by predicting the next action based on the current state, previous action and the desired return-to-go. While efforts have been made to extend DT-based architectures to offline MARL (Meng et al., 2021; Tseng et al., 2022), certain limitations persist. MADT (Meng et al., 2021) adapts DT independently for each agent in the multi-agent settings, failing to explicitly model cooperation between agents. (Tseng et al., 2022) adopts a similar approach and trains a centralized teacher policy to capture agent interactions, with individual agents learning through policy distillation. However, centralized transformers face scalability issues, requiring training on information from all agents. Additionally, these approaches inherit the inherent limitations of transformers, such as large model sizes, inefficient runtime inference, and restricted ability to capture long-range dependencies due to fixed window size constraints.

Structured State Space Sequence (S4) models (Gu et al., 2022) have recently been shown to outperform transformer-based models in single-agent offline RL tasks (Bar-David et al., 2023). Building on this success, we propose a sequence-learning-based offline MARL algorithm leveraging S4 variants. These models provide superior parameter efficiency compared to transformers, effectively capture longer temporal contexts, and enable constant-time inference over the quadratic time complexity of transformers with respect to the sequence length. Unlike previous works, such as MADT, which trains agents independently, our method explicitly models cooperation through a Sequentially Expanded MDP (SE-MDP) paradigm. In this framework, recently used in online MARL settings (Li et al., 2023), each decision step is divided into mini-steps, with agents acting sequentially based on their predecessors' actions. Unlike (Li et al., 2023), we enable limited communication, requiring each agent to access only its immediate predecessor's information, shared through the latent state representation of the S4 model. Utilizing this hidden state of the S4 module of the current agent, information on all its prior agents is efficiently passed down to the next agent, and gradients flow backward from the current agent through this shared memory to the previous agents during training. This design enables scalable training with constant memory communication overhead, unlike traditional communication-based MARL algorithms, where memory overhead increases quadratically with the number of agents. Additionally, this streamlined information-sharing mechanism helps mitigate non-stationarity issues during online fine-tuning. This form of training also shares similarities with the way information is passed between segments of long sequences in Recurrent Memory Transformer (RMT) (Bulatov et al., 2022).

The S4-based agents are trained directly on sequences or trajectories from the offline dataset in an efficient convolutional manner. The offline pre-trained models can be further used for sample-efficient online fine-tuning based on individual tuples instead of sequences leveraging the recurrent view of S4. We evaluate the performance of our developed algorithm, called Multi-Agent Decision S4 (MADS4), on the challenging offline MARL benchmarks of Multi-Robot Warehouse (RWARE) (Papoudakis et al., 2020) and StarCraft2 Multi-Agent Challenge (SMAC) (Samvelyan et al., 2019), where MADS4 achieves competitive or improved performance across many tasks over state-of-the-art offline RL-based and transformer-based baselines.

## 2 Related Work

**Offline Reinforcement Learning** Offline RL allows for policy learning based on pre-collected datasets without having access to active interactions with the environment (Levine et al., 2020), which is then directly used as the final policy or is used as a starting point for further improvement (Uchendu et al., 2023). This learning paradigm however results in severe distribution shift and extrapolation errors during policy evaluation on OOD samples not present in the offline dataset (Kumar et al., 2019; Fujimoto et al., 2019). Several approaches have been developed to mitigate this issue which typically involves various types of regularizations to be near the offline data distribution. Policy-based regularizations implicitly or explicitly constrain the policy to be close to the behavior policy of the dataset (Wu et al., 2019; Xu et al., 2021; Cheng et al., 2024; Li et al., 2022). Value-based regularizations aim to learn conservative value functions on OOD samples (Kumar et al., 2020; Kostrikov et al., 2021; Xu et al., 2022b). Other approaches involve including uncertainty (Wu et al., 2021; Bai et al., 2022) or penalizing OOD rewards (Yu et al., 2020).

In contrast to these regularization-based methods, Decision Transformer (DT) (Chen et al., 2021) takes a goal-conditioned supervised learning (GCSL) approach to formulate offline RL as a sequence modeling task and outperforms many state-of-the-art offline RL algorithms. Following the success of this training

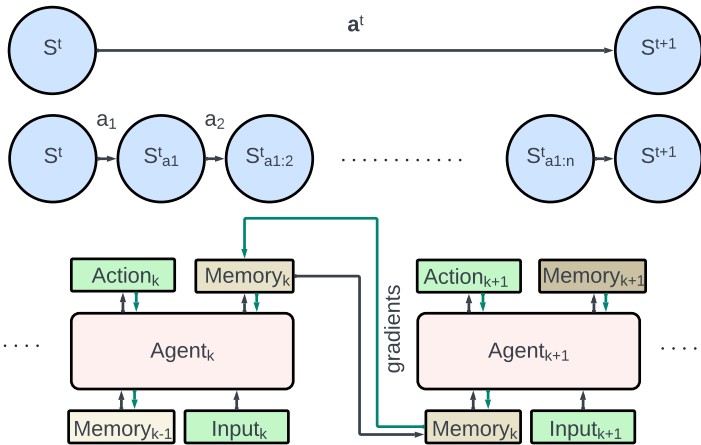

Figure 1: Multi-Agent MDP (MMDP) is restructured into a Sequentially-Expanded MDP (SE-MDP) where the multi-agent state transition at each timestep is decomposed into $n$ intermediate states. In this framework, in the forward pass (black lines), each agent processes its input alongside information received from its preceding agent, then takes an action and passes updated information to the next agent in the sequence. During training, the gradient flows backward (green lines), enabling earlier agents to receive updates based on the information passed by later agents.

regime, Decision S4 (Bar-David et al., 2023) proposes using S4 model variants for higher parameter efficiency, capturing longer sequences and faster inference.

**Offline MARL**  Extending single-agent RL methods to multi-agent settings presents significant challenges due to the exponential growth of the joint state-action space. Most MARL algorithms adopt the Centralized Training with Decentralized Execution (CTDE) paradigm. In CTDE, global information is shared during training, and local, decoupled policies are used for execution (Oliehoek et al., 2008; Sunehag et al., 2017; Rashid et al., 2020; Son et al., 2019; Wang et al., 2020; Foerster et al., 2017; Lowe et al., 2017; Yu et al., 2021). Recently, offline RL-based MARL algorithms have emerged, typically applying regularizations on local policies or value functions (Yang et al., 2021; Jiang & Lu, 2023; Pan et al., 2022). On the other hand, (Meng et al., 2021) extends the Decision Transformer (DT) (Chen et al., 2021) to a multi-agent setting, where agents are trained independently by sharing weights within a goal-conditioned supervised learning framework. However, these algorithms do not provide guarantees of global-level regularizations and fail to explicitly or implicitly learn cooperative behavior. Only a few recent works have tried to tackle these limitations. For example, (Wang et al., 2024) uses an implicit global to-local regularization, and (Tseng et al., 2022) uses knowledge distillation to distill cooperation in the local policies.

**S4**  S4(Gu et al., 2022; 2021) and their variants (Gupta et al., 2022; Smith et al., 2022), which are developed on time-invariant linear state space layers, have outperformed transformers in capturing long-range contexts. These models require far fewer parameters and have constant time inference; hence, they have been suitably utilized in reinforcement learning domains in single-agent learning (Bar-David et al., 2023) and in-context learning (Lu et al., 2024). Commonly, the model uses the convolutional mode for efficient parallelizable training (where the whole input sequence is seen ahead of time) and switched into a recurrent mode for efficient autoregressive inference (where the inputs are seen one timestep at a time).

# 3 Methodology

## 3.1 Problem Formulation

In this section, we present our cooperative multi-agent decision-making approach based on sequence learning with state-space layers. We design our approach based on the Multi-agent Markov Decision Process (MMDP) framework (Littman, 1994). An MMDP over $n$ agents is defined by the tuple $\mathcal{G} = \langle \mathcal{S}, \mathcal{A}, P, r, \gamma \rangle$, where $\mathcal{S}$ is the set of global states, and $\mathcal{A} = \mathcal{A}_1 \times \cdots \times \mathcal{A}_n$ is the joint action space. At each timestep, the environment transitions from state $s \in \mathcal{S}$ to $s' \in \mathcal{S}$ under the joint action $\mathbf{a} = [a_1, \ldots, a_n]$, following the transition function $P(s' \mid s, \mathbf{a})$, and returns a global reward $r(s, \mathbf{a})$. The goal is to learn a joint policy $\pi(\mathbf{a} \mid s)$ that maximizes the expected return $\mathbb{E}_\pi \left[ \sum_{t=0}^{\infty} \gamma^t r(s^t, \mathbf{a}^t) \right]$.

## 3.2 Sequentially Expanded MDP

In this work, the Multi-agent Markov Decision Process (MMDP) is transformed into a Sequentially Expanded Markov Decision Process (SE-MDP), where each timestep is divided into $n$ mini timesteps and a multi-agent decision by $n$ agents is expanded into a sequence of $n$ individual decisions, with only one agent acting during each mini-timestep. Thus, a single-step transition in the original MMDP $(s^t, \mathbf{a}^t, s^{t+1})$ resulting in a shared reward $r(s^t, \mathbf{a}^t)$ is composed of a sequence of $n$ intermediate transitions, which in turn result in the same shared reward $r(s^t, \mathbf{a}^t)$, as shown in Figure 1.

$$(s^t, \mathbf{a}^t, s^{t+1}) = \{(s^t, a_1^t, s_{a_1}^t), (s_{a_1}^t, a_2^t, s_{a_{1:2}}^t), ..., (s_{a_{1:n-1}}^t, a_n^t, s_{a_{1:n}}^t = s^{t+1})\} \tag{1}$$

Within this framework, at each timestep, an agent's action is based on information passed by the immediately preceding agent in the sequence. This creates a bidirectional dependency between the agents as shown in Figure 1: in the forward direction, an agent's action is influenced by the actions of its predecessors, while in the backward direction, gradients can propagate from the current agent back to the previous agents.

## 3.3 Sequence-based Reinforcement Learning

Our approach follows the offline RL paradigm using sequence modeling, where the RL problem is reframed as a supervised learning task by predicting actions in an autoregressive manner—typically conditioned on the current state, past actions, and a target return-to-go. The return-to-go at timestep $t$, denoted $R_t$, is defined as the cumulative future reward from that point until the end of the decision horizon $T$: $R_t = \sum_{i=t}^{T} r_i$. Conditioning on return-to-go rather than immediate rewards enables the model to associate state-action sequences with desirable long-term outcomes (Chen et al., 2021). We extend this framework to the multi-agent setting, where the state, action, and reward of the $i^{th}$ agent at each timestep are denoted as $s_i, a_i, r_i$, and its trajectories $\tau : (s_0, a_0, r_0, s_1, a_1, r_1, ..., s_T, a_T, r_T)$ consist of sequences of state, action, and reward tuples. To align with return-conditioned learning, these trajectories are restructured as $\tau = (R_0, s_0, a_0, R_1, s_1, a_1, \ldots, R_T, s_T, a_T)$, where $R_i$ is the return-to-go from timestep $i$ onward.

## 3.4 S4-based Agent

This section provides background on State Space Models (SSMs) and Structured State Space Models (S4), which form the basis for designing individual agents in our framework.

SSMs are a classical framework for modeling sequential data through latent dynamical systems. At each timestep $t$, the model receives an input $u(t)$, updates an internal memory or hidden state $x(t)$, and generates an output $y(t)$ using a first-order linear time-invariant differential equation:

$$\begin{aligned} \dot{x}(t) &= \mathbf{A}x(t) + \mathbf{B}u(t) \\ y(t) &= \mathbf{C}x(t) + \mathbf{D}u(t) \end{aligned} \tag{2}$$

SSMs typically operate on continuous time sequences where $A, B, C, D$ are parameter matrices of appropriate dimensions. To apply SSMs in discrete-time domains, such as typical reinforcement learning settings, the continuous-time formulation is discretized with a fixed step size $\Delta$ using schemes like the bilinear transform (Tustin, 1947), yielding the discretized linear recurrence:

$$
\begin{aligned}
x_k &= \bar{A} x_{k-1} + \bar{B} u_k \\
y_k &= \bar{C} x_k + \bar{D} u_k
\end{aligned}
\tag{3}
$$

where the discretized matrices $\bar{A}, \bar{B}, \bar{C}, \bar{D}$ are functions of the continuous-time parameters and the step size $\Delta$.

S4 models (Gu et al., 2022) build upon this formulation by introducing efficient parameterizations to enable stable and scalable modeling of long sequences. S4 leverages special initialization strategies such as HiPPO (Gu et al., 2020) to preserve long-range dependencies, and its structured formulation allows it to be integrated effectively with deep learning models for sequence modeling tasks. Various techniques have since been developed to improve the model's performance, stability, and training efficiency (Gu et al., 2021; Gupta et al., 2022). Similar to these works, $D$ is represented here by a skip connection.

Importantly, since Eq. 3 is linear and time-invariant, the output sequence $y$ can be computed directly in parallel based on the input sequence $u$ via non-circular convolution with a kernel $\bar{K}$:

$$
\begin{aligned}
y_k &= \bar{C} \bar{A}^k \bar{B} u_0 + \bar{C} \bar{A}^{k-1} \bar{B} u_1 + \cdots + \bar{C} \bar{A} \bar{B} u_{k-1} + \bar{C} \bar{B} u_k \\
y &= \bar{K} * u
\end{aligned}
\tag{4}
$$

where $\bar{K}$ is a function of $\bar{A}, \bar{B}, \bar{C}, \bar{D}$ and context length $L$ which is pre-fixed during training. This convolution can be computed efficiently across all time steps, allowing for parallelizable training. The recurrent view of the SSM also allows for faster inference with low memory. This is a key advantage over transformers, which makes the use of SSMs very effective in reinforcement learning settings, which require faster inference for the collection of online interactions with the environment. Additional details can be found in Appendix A.

In this work, each agent is modeled using an S4-based architecture that processes sequences of past information and target returns-to-go to predict the next action. We then extend this design to the multi-agent setting by enabling cooperation through information sharing, as detailed below.

### 3.5 Information Sharing with Limited Communication

To facilitate scalable cooperation among S4-based agents, we introduce a communication mechanism restricted to consecutive agents within the SE-MDP sequence. Specifically, each agent's memory information—captured by the hidden state of its S4 module—is passed to the next agent in the sequence. As a result, the hidden state of an agent in the sequence implicitly encodes information from all preceding agents. This design induces a bidirectional flow of information and dependency across agents. At each timestep $t$, a projection of the previous agent's latent state, $h_{i-1}^t$, is provided as an additional input to the next agent, alongside its own input $\hat{u}_i^t$, thereby influencing both its action $a_i^t$ and updated memory state $h_i^t$:

$$
a_i^t, h_i^t = \pi_i(\hat{u}_i^t, h_{i-1}^t; \theta_i)
\tag{5}
$$

During training, gradients flow backward through the shared latent states, enabling the entire system to learn cooperative strategies:

$$
\frac{\partial J}{\partial \theta_i} = \frac{\partial J}{\partial a_i^t} \cdot \frac{\partial a_i^t}{\partial \theta_i} + \frac{\partial J}{\partial a_{i+1}^t} \cdot \frac{\partial a_{i+1}^t}{\partial h_i^t} \cdot \frac{\partial h_i^t}{\partial \theta_i}.
\tag{6}
$$

where $J$ represents the supervised loss function computed across all agents in the system. This sequential flow of information eliminates the need for an agent to communicate with more than one peer or identify useful collaborators, a challenge that grows with the number of agents. In contrast to typical communication-based MARL algorithms, which scale poorly due to the quadratic growth in memory requirements during training and execution, our mechanism is highly efficient, requiring only constant memory per agent.

We first adapt Decision S4 (DS4) for each agent with parameter sharing, similar to the Multi-Agent Decision Transformer (MADT) (Meng et al., 2021). Unlike MADT, however, the Multi-Agent Decision S4 (MADS4) is trained in a sequentially dependent manner, where agents can share accumulated memory information with the next agent in the sequence. The offline version of MADS4, trained on pre-collected trajectories, is detailed in the next section. These pre-trained models can be further fine-tuned in an on-policy setting using MAPPO (Yu et al., 2021).

## 4 Multi-Agent Decision S4 (MADS4)

### 4.1 MADS4: Offline Training

**Input formulation** In the offline training setup of MADS4, each agent is trained over trajectories consisting of sequences of previously seen observations, its own previously executed actions, the latent state representation of its preceding agent, and returns to go from the current time step. Similar to MADT, the state of each agent at each time step $s_i^t$ is composed of global environment state $s_{gi}^t$ and its local observation $o_i^t$. However, our model performs very similarly without using the global state information in the input, as shown in Appendix C.2. Thus, a trajectory for the $i^{th}$ agent, which is taken as input to the S4-based model, consists of the following:

$$\tau_i = (u^1, u^2, ..., u^T) \quad \text{where} \quad u^t = \{R^t, s_{gi}^t, o_i^t, a_i^{t-1}, h_{i-1}^t\} \tag{7}$$

where $R^t$ is the returns-to-go from current time step $t$, $s_{gi}^t$ is the current shared global state, $o_i^t$ is the current local observation, $a_i^{t-1}$ is the previously executed action of the

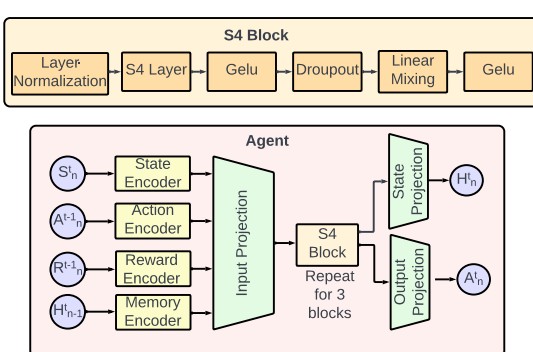

Figure 2: MADS4 Actor Network. In addition to encoding the state, action, and reward, the memory state from the previous agent $(H_{n-1})$ is processed through a memory encoder. The updated S4 states and outputs are then projected via separate heads.

$i^{th}$ agent, $h_{i-1}^t$ is the current hidden state representation of the preceding agent. The model is trained to predict actions (action logits) at time step $t$ in an autoregressive manner based on the data seen so far. The output of the S4-based model is used as the action probabilities which are sampled after applying the action availability masks.

$$\hat{a}_i^t = \arg\max_a P(a_i^t | \tau_i^{\leq t}; \theta) \tag{8}$$

where $\theta$ are the parameters of the MADS4. In this work, parameter sharing is allowed across the agents for training stability, and thus, essentially, a single model is trained, which takes into account different inputs for different agents along with their specific one-hot agent IDs.

**Network architecture and training** The MADS4 architecture consists of three key components, as illustrated in Figure 2: (i) Input, state, and output projection layers: each of these consists of a fully connected layer followed by ReLU activations; (ii) Input encoder layers: These layers handle states, actions, and rewards/returns, each implemented as fully connected linear layers; (iii) Sequence modeling component: This component consists of stacked S4 blocks, where each block consists of Batch Normalization layer followed by S4 layer, linear mixing layer with GELU activation, and a dropout layer. We employed various kernels

---

**Algorithm 1** MADS4-Offline Training

---

**Input:** Offline dataset $\mathcal{D} : \{\tau_i : \langle s_{gi}^t, o_i^t, a_i^t, v_i^t, d_i^t, R_i^t \rangle_{t=1}^T\}_{i=1}^n$, where $n$ is the number of agents, and $v_i^t$ denotes the available actions for the $i^{th}$ agent at time $t$; $d_i^t$ denotes the done signal for an episode
**Initialize** $\alpha$ as the learning rate, $L$ as the context length
**Initialize** $\theta$ for the S4 models based on HIPPO initialization

1: **for** $i = 1 : n$ **do**                                      ▷ Iterate over each agent
2:     From $\tau_i$ and $h_{i-1}^t$, create $X_i : \{R_i^t, s_{gi}^t, o_i^t, a_i^{t-1}, h_{i-1}^t\}_{t=1}^T$, where $h_{i-1}^t$ is the latent state representation of the previous agent, assuming $h_{-1}^t = \mathbf{0}$
3:     Zero-pad the trajectory to a constant length $L$ when $d_i^t$ is true          ▷ Pad when the agent is done
4:     Compute action output sequence $\hat{a}_i = \{\hat{a}_i^1, \ldots, \hat{a}_i^T\}$ and latent state projections $h_i = \{h_i^1, \ldots, h_i^T\}$
5:     **for** $t = 1 : T$ **do**                                ▷ Loss calculation over time steps
6:         Mask illegal actions via $P(\hat{a}_{ij}^t | \tau_i^{\leq t}; \theta) = 0$ if $v_{ij}^t$ is False, where $j$ is the unavailable action index
7:         Predict the action $\hat{a}_i^t = \arg\max_j P(\hat{a}_{ij} | \tau_i^{\leq t}; \theta)$
8:     **end for**
9:     Update $\theta$:

$$\theta \leftarrow \arg\max_\theta \frac{1}{T} \sum_{t=1}^T P(a_i^t) \log P(\hat{a}_i^t | \tau_i^{\leq t}; \theta)$$

10: **end for**
**Return:** $\theta$

---

for S4, with the "Normal Plus Low Rank" kernel initialized using HIPPO, achieving the best performance. In all experiments, we set the input channel size to $H = 96$ and the S4 state size to $N = 96$. The effect of varying state and input sizes is shown in Section 5.4.

In the offline setting, the S4 model is trained efficiently using the convolutional view on entire trajectories sampled randomly from the offline dataset. The trajectories are zero-padded to a constant context length. Unlike transformers, which face limitations on context length due to the expensive quadratic time and space complexity of self-attention, S4-based models can be trained on complete trajectories that are often much longer than those typically used for transformers in most environments. The impact of truncating trajectory lengths has significant implications for model performance, as shown in Section 5.4. Actions are predicted based on the action logit outputs of the model, and the model is trained based on loss computed using cross-entropy between the true action labels and the predicted actions.

### 4.2   MADS4: On-policy Fine-tuning

For online fine-tuning, the offline pre-trained agent is used to interact with the online environment and is further updated based on an on-policy training scheme. The agent interacts with the environment while creating the buffer, which stores the local observations and actions of the individual agents, shared global states of the environment, rewards, returns-to-go, and also the latent states of the S4 modules of the agents. Within the well-known MAPPO-based (Yu et al., 2021) actor-critic framework, the offline-pretrained S4-based model is used as actor networks of the agents which predict actions via the recurrent view based on latent states and other inputs as used in the pretraining stage. The critic network is conditioned on both the global states of the environment as well as the encoded latent S4 states to evaluate the state value function.

**Network architecture and training**   The pre-trained model is loaded as the actor network, which predicts action probabilities and next states as:

$$p_i^t, h_i^t = \pi(u_i^t, h_i^{t-1}; \theta) \quad \text{where} \quad u_i^t = \{R_i^t, s_{gi}^t, o_i^t, a_i^{t-1}, h_{i-1}^t\} \tag{9}$$

---

**Algorithm 2** MADS4: On-policy Fine-tuning

---

1: Copy the model weights $\theta$ to the actor or policy network $\pi : \pi(u_i)$, where $u_i = \{R_i^t, s_{gi}^t, o_i^t, a_i^{t-1}, h_{i-1}^t\}$; Initialize $\phi$, the parameters for critic $V$
2: Set learning rates $\alpha_\pi, \alpha_V$ for actor and critic
3: **for** iterations $= 1, M$ **do**
4:     Set data buffer $D = \{\}$
5:     **for** $i = 1$ to *batch_size* **do**
6:         $\tau = []$                                                   $\triangleright$ Empty list
7:         Initialize $h_0^{(1)}, \ldots, h_0^{(n)}$ actor S4 states
8:         **for** each timestep $t$ in the environment **do**
9:             **for** agent $= 1 : n$ **do**
10:                 $p_i^t, h_i^t = \pi(u_i^t, h_{i-1}^t; \theta)$
11:                 Sample $a_i^t \sim p_i^t$
12:             **end for**
13:             Execute actions $\boldsymbol{a}^t$, observe $r^t$, $\boldsymbol{s}_g^{t+1}$, $\boldsymbol{o}^{t+1}$
14:             $\tau += [\boldsymbol{s}_t, \boldsymbol{o}_t, \boldsymbol{h}_t, \boldsymbol{a_t}, r_t, R_t, \boldsymbol{s}_{t+1}, \boldsymbol{o}_{t+1}]$
15:         **end for**
16:         Calculate advantage $A^t$ via GAE on $\tau$ and store $\tau$ with $A^t$ in the buffer $D$
17:     **end for**
18:     **for** $k = 1, \ldots, K$ training steps **do**
19:         Sample batch from replay-buffer $B = \{(s_g^t, o^t, a^{t-1}, R^{t-1}, h^{t-1}, s_g^{t+1}, o^{t+1}, a^t, h^t, r^t, R^t, A^t)\} \subset D$ for each agent
20:         Calculate Bellman target estimate: $y = r^t + \gamma V(s^{t+1}, h^t)$
21:         Update critic: $\phi_V = \phi_V - \alpha_V \nabla_{\phi_V}(V(s^t, h^{t-1}) - y)^2$
22:         If actor freezing is over, update actor:

$$\theta_\pi \leftarrow \arg\max_{\theta_\pi} \mathbb{E}_{s \sim \rho_{\theta_{\text{old}}}, a \sim \pi_{\theta_{\text{old}}}} \left[ \text{clip}(w, 1-\epsilon, 1+\epsilon) A^t \right]$$

        where the importance weight $w = \frac{\pi_\theta(a_i|o_i)}{\pi_{\theta_{\text{old}}}(a_i|o_i)}$
23:     **end for**
24: **end for**=0

---

The critic network is parameterized by fully connected layers with ReLU activations, which take the shared global state of the environment and encoded latent states and evaluate the value function, which is used to update the S4-based actor parameters ($\theta$) using the policy gradient theorem. For more stable training. the actor network is kept frozen initially, and the critic is solely trained on the recorded data collected using the pre-trained actor. After sufficient training of the critic, the actor and critics are simultaneously trained. During exploration, the desired returns-to-go is set at 10% higher than the current model's highest return. Additional details on the experimental setup and training are provided in Appendix A.4.

## 5 Experiments

### 5.1 Datasets and Baselines

We evaluate the performance of MADS4 on challenging cooperative MARL benchmarks of Multi-Robot Warehouse (RWARE) (Papoudakis et al., 2020) and StarCraft2 Multi-Agent Challenge (SMAC) (Samvelyan et al., 2019). The offline datasets in the RWARE domain are obtained from (Matsunaga et al., 2023), which consists of diverse trajectories collected by training Multi-Agent Transformer (MAT) (Wen et al., 2022). The RWARE datasets consist of expert policies trained on 2 maps (tiny and small) with different numbers of agents. For the SMAC domain, the datasets provided by (Meng et al., 2021) have been used, which consists of trajectories collected with online trained MAPPO agents. The datasets consist of three trained quality levels of the agents, good, medium, and poor, tested on the different SMAC maps. For this work, we chose

Table 1: Average returns and standard deviations over 5 random seeds on the Warehouse domain.

| Method | Tiny (11x11) | | | Small (11x20) | | |
|---|---|---|---|---|---|---|
| | (N = 2) | (N = 4) | (N = 6) | (N = 2) | (N = 4) | (N = 6) |
| BC | 8.80 ± 0.25 | 11.12 ± 0.19 | 14.06 ± 0.32 | 5.54 ± 0.06 | 7.88 ± 0.14 | 8.90 ± 0.13 |
| ICQ | 9.38 ± 0.75 | 12.13 ± 0.44 | 14.59 ± 0.16 | 5.43 ± 0.19 | 7.93 ± 0.19 | 8.87 ± 0.22 |
| OMAR | 6.77 ± 0.64 | 14.39 ± 0.91 | 16.13 ± 1.21 | 4.40 ± 0.34 | 7.12 ± 0.38 | 8.41 ± 0.49 |
| MADTKD | 6.24 ± 0.60 | 9.90 ± 0.21 | 13.06 ± 0.19 | 3.65 ± 0.34 | 6.85 ± 0.36 | 7.85 ± 0.52 |
| OptiDICE | 8.70 ± 0.06 | 11.13 ± 0.44 | 14.02 ± 0.36 | 4.84 ± 0.32 | 7.68 ± 0.09 | 8.47 ± 0.26 |
| AlberDICE | **11.15 ± 0.35** | 13.11 ± 0.32 | 15.72 ± 0.36 | 5.97 ± 0.11 | 8.18 ± 0.19 | 9.65 ± 0.13 |
| **MADS4 (ours)** | **11.79 ± 0.61** | **15.52 ± 0.20** | **17.29 ± 0.76** | **6.58 ± 0.28** | **9.47 ± 0.15** | **10.87 ± 0.55** |

four representative maps consisting of two hard (5m vs. 6m, 2c vs. 64zg) and two super hard (6h vs. 8z, corridor) maps for evaluating MADS4. Additional statistics on the offline datasets can be found in Appendix B.

For comparisons on both domains, we compare with several recent offline MARL algorithms from the paradigms of both offline reinforcement learning and sequence-based supervised learning. The offline RL baselines considered for comparison are Behaviour Cloning (BC) (Fujimoto et al., 2019), OptiDICE (Lee et al., 2021), AlberDICE (Matsunaga et al., 2023), ICQ (Yang et al., 2021), OMAR(Pan et al., 2022) and OMIGA(Wang et al., 2024). The sequence-based learning algorithms considered in this work include MADT (Meng et al., 2021) and MADTKD (Tseng et al., 2022), which are based on transformers. MADT policies do not involve any cooperation during learning, whereas MADTKD incorporates a degree of cooperation distilled into the agents from the centralized teacher model.

## 5.2 Offline Training

Here, we compare the performance of offline trained MADS4, where the agents share information in the form of their latent state projections. The trained agents are deployed on the online RWARE and StarCraft2 environments for evaluation. Tables 1 and 2 show the mean and standard deviation of average returns in RWARE and SMAC domains, respectively, evaluated over 30 episodes and 5 different training seeds. During evaluation, the desired returns-to-go is set at 10% higher than the highest returns encountered in the offline datasets.

**RWARE** environment is a warehouse simulation consisting of agents moving and delivering goods to workstations in partially observable settings while avoiding collisions. This domain poses challenges due to high-dimensional observations and the need for strong cooperation, especially in high-density settings where agents must navigate narrow passages. In this domain, MADS4 outperforms all baselines across the maps, with a larger performance gap on the small and tiny maps involving 6 agents, where tight coordination is crucial to avoid collisions in confined spaces. MADS4 also outperforms transformer-based baselines like MADTKD, likely due to the long trajectories in the RWARE datasets (up to 500 timesteps), which are often truncated to reduce transformer training costs. In contrast, MADS4 processes full trajectories, capturing longer contexts with fewer parameters.

In the **SMAC** domain, MADS4 demonstrates consistent performance that is similar to or better than the considered baselines across all studied maps. Notably, the model outperforms all baselines in the hard and superhard maps, specifically in the 2c vs. 64zg and 6h vs. 8z scenarios.

## 5.3 On-policy Fine-tuning

We evaluate whether the performance of offline pre-trained models can be enhanced through on-policy fine-tuning. During this phase, MADS4 interacts with the environment, collecting trajectories that are stored in a buffer and used to update the S4-based models via recurrence. As shown in Figure 3, on-policy training builds upon and improves the offline pretraining results. Furthermore, on-policy training without pretraining consistently results in sub-optimal performance across all tasks, underscoring the importance of pretraining for achieving superior results.

Table 2: Average returns and standard deviations over 5 random seeds on the SMAC domain.

| SMAC Map | Data | RL-based | | | Sequence-based | |
| --- | --- | --- | --- | --- | --- | --- |
| | | ICQ | OMAR | OMIGA | MADT | **MADS4 (ours)** |
| 5m vs 6m (H) | G | $7.87 \pm 0.30$ | $7.40 \pm 0.63$ | $\mathbf{8.25 \pm 0.37}$ | $8.15 \pm 0.63$ | $8.00 \pm 0.45$ |
| | M | $7.77 \pm 0.30$ | $7.08 \pm 0.51$ | $\mathbf{7.92 \pm 0.57}$ | $7.80 \pm 0.56$ | $7.85 \pm 0.57$ |
| | P | $7.26 \pm 0.19$ | $7.27 \pm 0.42$ | $7.52 \pm 0.21$ | $7.23 \pm 0.48$ | $\mathbf{7.67 \pm 0.15}$ |
| 2c vs 64zg (H) | G | $18.82 \pm 0.17$ | $17.27 \pm 0.78$ | $19.15 \pm 0.32$ | $18.90 \pm 0.78$ | $\mathbf{19.40 \pm 0.55}$ |
| | M | $15.57 \pm 0.61$ | $10.20 \pm 0.20$ | $16.03 \pm 0.19$ | $16.92 \pm 0.20$ | $\mathbf{17.27 \pm 0.15}$ |
| | P | $12.56 \pm 0.18$ | $11.33 \pm 0.50$ | $13.02 \pm 0.66$ | $13.33 \pm 0.50$ | $\mathbf{14.67 \pm 0.32}$ |
| 6h vs 8z (SH) | G | $11.81 \pm 0.12$ | $9.85 \pm 0.28$ | $12.54 \pm 0.21$ | $12.55 \pm 0.67$ | $\mathbf{12.75 \pm 0.15}$ |
| | M | $11.13 \pm 0.33$ | $10.36 \pm 0.16$ | $12.31 \pm 0.22$ | $12.36 \pm 0.16$ | $\mathbf{12.57 \pm 0.25}$ |
| | P | $10.55 \pm 0.10$ | $10.63 \pm 0.25$ | $11.67 \pm 0.19$ | $11.63 \pm 0.25$ | $\mathbf{11.89 \pm 0.43}$ |
| corridor (SH) | G | $15.54 \pm 1.12$ | $6.74 \pm 0.69$ | $15.88 \pm 0.89$ | $\mathbf{17.81 \pm 1.14}$ | $16.02 \pm 0.97$ |
| | M | $11.30 \pm 1.57$ | $7.26 \pm 0.71$ | $11.66 \pm 1.30$ | $12.75 \pm 1.18$ | $\mathbf{12.80 \pm 1.12}$ |
| | P | $4.47 \pm 0.33$ | $4.28 \pm 0.49$ | $5.61 \pm 0.35$ | $\mathbf{8.76 \pm 0.49}$ | $8.57 \pm 0.54$ |

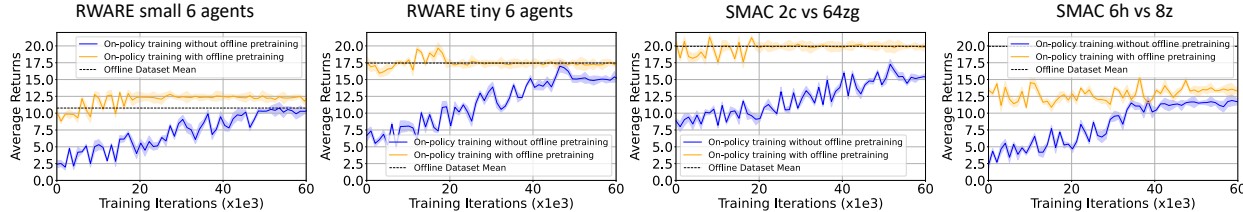

Figure 3: Training curves of on-policy training of MADS4 with and without offline pretraining. Mean and standard deviations of average returns are plotted over 5 independent runs.

However, prolonged on-policy training can sometimes degrade the performance of pre-trained models, as shown in Appendix C.1. This degradation likely arises from the inherent instability of training S4 modules in a recurrent setup, compared to the more stable convolution-based operations employed during offline pretraining, leading to error accumulation. To address this, we mitigate the issue by freezing the S4 kernel parameter $A$, which governs state-to-state transitions independent of inputs, and fine-tuning only the input-dependent parameters $B$ and $C$.

### 5.4 Ablation Study and Hyperparameter Analysis

**Effect of sharing information** The sharing of information between agents leads to significantly improved cooperative behavior, as reflected in the higher average rewards shown in Figure 4. This performance boost is particularly pronounced in more complex tasks that involve a greater number of agents and demand precise coordination. Importantly, this method of sharing information is scalable, where an agent only needs to communicate with the next agent, minimizing overhead while ensuring efficient coordination. Information can be shared in multiple forms, namely by passing the action logit outputs from the S4 model, the latent state representations, or a combination of both from the preceding agent. A more detailed analysis of how different types of shared information impact performance is provided in Appendix A.2.

**Effect of order of agents** To assess the impact of agent ordering on MADS4's performance, we compared two training settings: (1) Random Order, where the agent order is randomly shuffled during training, and (2) Fixed Order, where agents are trained in the same sorted order as in the offline dataset. Figure 5(a) shows similar performance, demonstrating that MADS4 is robust to agent ordering within the SE-MDP framework. Nonetheless, we recommend using a random order during training to avoid introducing potential biases into the learning process. Notably, our approach only requires each agent to communicate with one unique peer

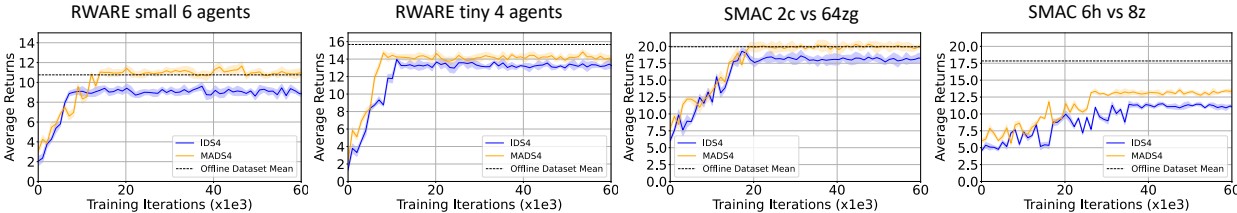

Figure 4: Training curves of MADS4 with information sharing between consecutive agents and IDS4 where agents are trained independently. Mean and standard deviations of average returns are plotted over 5 independent runs.

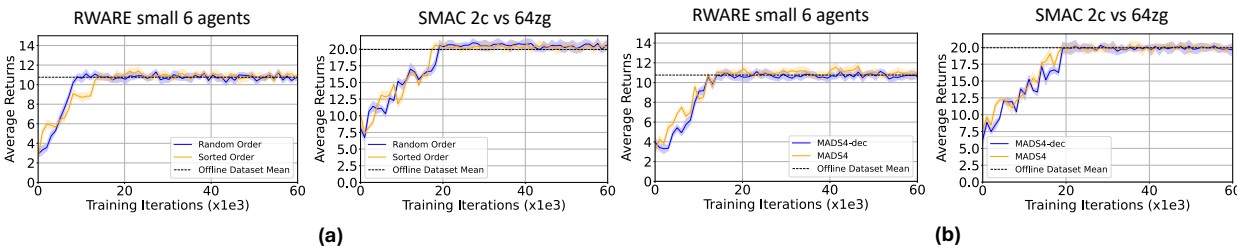

Figure 5: (a) The effect of having a shuffled random order vs. a fixed sorted order of the agents in the SE-MDP framework on the RWARE domain in the small 6 agents scenario and SMAC domain in the 2c vs. 64zg map. (b) The comparison of the performance of MADS4 vs. MADS4-dec (decentralized MADS4) on RWARE small map with 6 agents and SMAC map 2c vs 64 zg.

which can be selected randomly to ensure that every agent's information is passed across the network without the need for any centralized optimization or sophisticated coordination.

**MADS4 in decentralized setting**   To adapt MADS4 for a decentralized setting, where agents act in parallel, we leverage the hidden state information of each agent from the previous timestep as a proxy for the current timestep. When decisions are made at the current timestep, all decisions from the previous timestep will already be finalized. As a result, the memory information of all agents is readily available for use. By utilizing the memory information from the previous timestep, agents can make decisions without relying on sequential dependencies during the current timestep. Since memory accumulates over multiple timesteps, relying on the previous timestep's information does not compromise performance, as demonstrated in Figure 5(b). This modification enables our algorithm to function effectively in decentralized policy settings without performance degradation.

**S4 model size parameters**   We analyze the impact of the S4 model size parameters, specifically the number of input channels ($H$) and the latent state size ($N$), on the model performance, as shown in Table 3. We compare the total number of parameters against the 1.8 million parameters reported for MADTKD in (Tseng et al., 2022). Our biggest model with $N$=96 and $H$=96 was used in all our experiments, which consists of about 200k parameters.

**Effect of context length**   The context length used for pretraining significantly impacts performance, which is also evident for transformer-based models. In our experiments, we used the maximum trajectory lengths encountered in the offline datasets for pretraining. Representative results are shown in Figure 6, which illustrates the effects of truncating the trajectory lengths to various percentages of the maximum length in the offline dataset for the SMAC map 2c vs 64zg.

Table 3: Average returns for different MADS4 model configurations on the RWARE Small map. Each of the models is denoted by (i) $N$, the $S4$ state size, and (ii) $H$, the number of input/output channels. For parameter efficiency, we report the percentage of parameters used relative to MADTKD.

| Environments | (N=96,H=96) | (N=64,H=64) | (N=32,H=32) | (N=64,H=96) | (N=96,H=64) | (N=32,H=64) | **MADTKD** |
|---|---|---|---|---|---|---|---|
| 2 agents | 6.58 | 6.21 | 5.53 | 6.53 | 6.25 | 5.87 | 3.65 |
| 4 agents | 9.47 | 8.86 | 8.57 | 9.15 | 8.88 | 8.64 | 6.85 |
| 6 agents | 10.87 | 10.31 | 9.55 | 10.76 | 9.97 | 9.85 | 7.85 |
| % Parameters (Ours) | 100 | 60 | 40 | 81 | 82 | 55 | 100 |
| % Parameters (MADTKD) | 12 | 7 | 5 | 8 | 8 | 6 | 100 |

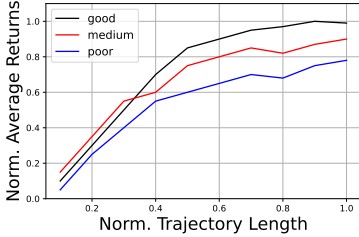

Figure 6: The effect of truncating the trajectory length during training. The average returns are normalized with the maximum returns encountered in the offline dataset.

## 6 Conclusions

In this work, we demonstrate the effectiveness of S4-based models in outperforming transformer-based architectures for sequence-to-sequence offline multi-agent reinforcement learning (MARL) tasks. By structuring agent interactions within the SE-MDP framework and limiting communication to the exchange of information between unique, arbitrarily chosen agent pairs, MADS4 enables more scalable and efficient cooperation. This contrasts with state-of-the-art offline RL and centralized transformer-based models, which require complete access to all agents' information during training, leading to scalability challenges. Additionally, MADS4 offers a low-latency and lightweight model that can be trained more efficiently than transformers and fine-tuned online using recurrent computations.

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

# A  Additional Background, Experimental Setup and Training Details

## A.1  S4 Layer

S4(Gu et al., 2022) layer is a variant of linear and time-invariant (LTI) state-space model (SSM)(Gu et al., 2021) which adopts the HIPPO (Gu et al., 2020)-based initializations in order to better capture longer contexts, and proposes efficient ways for kernel computations and parallel training.

### A.1.1  Recurrent View

Given an input scalar function $u(t) : \mathbb{R} \to \mathbb{R}$, the continuous LTI SSM is defined by the following first-order differential equation:

$$\dot{x}(t) = Ax(t) + Bu(t), \quad y(t) = Cx(t) + Du(t) \tag{10}$$

The model maps the input stream $u(t)$ to $y(t)$. It was shown that initializing $A$ by the HIPPO matrix (Gu et al., 2020) grants the state-space model (SSM) the ability to capture long-range dependencies. Similar to previous works (Gu et al., 2022; Gupta et al., 2022), $D$ is replaced by parameter-based skip-connection and is omitted from the SSM by assuming $D = 0$.

This SSM operates on continuous sequences, and it is discretized by a step size $\Delta$ to operate on discrete sequences. Let the discretization matrices be $\bar{A}, \bar{B}, \bar{C}$:

$$\bar{A} = (I - \Delta A/2)^{-1}(I + \Delta A/2), \quad \bar{B} = (I - \Delta A/2)^{-1}\Delta B, \quad \bar{C} = C \tag{11}$$

These matrices allow us to rewrite Eq. 10:

$$x_k = \bar{A}x_{k-1} + \bar{B}u_k, \quad y_k = \bar{C}x_k \tag{12}$$

Using the recurrent Eq.12, SSM asymptotically allows for constant $O(1)$ time and memory inference for each token/ timestep, as compared to $O(L^2)$ inference for transformers. SSM can be interpreted as a linear RNN in which $\bar{A}$ is the state-transition matrix, and $\bar{B}, \bar{C}$ are the input and output matrices. Thus, it essentially requires $O(L)$ training, $L$ being the sequence length, as compared to $O(L^2)$ (parallelizable) training complexity for transformers.

### A.1.2  Convolutional View

The recurrent SSM view is not practical for training over long sequences, as the training cannot be parallelized across the sequence dimension and results in instabilities from vanishing gradient issues. However, the LTI SSM can be rewritten as a convolution, which allows for efficient parallelizable training. The S4 convolutional view is obtained as follows:

Given a sequence of scalars $u = (u_0, u_1, ..., u_{L-1})$ of length $L$, the S4 recurrent view can be unrolled to the following closed form:

$$\forall i \in [L-1] : x_i \in \mathbb{R}^N, \quad x_0 = \bar{B}u_0, \quad x_1 = \bar{A}\bar{B}u_0 + \bar{B}u_1, \quad ..., \quad x_{L-1} = \sum_{i=0}^{L-1} \bar{A}^{L-1-i}\bar{B}u_i$$

$$y_i \in \mathbb{R}, \quad y_0 = \bar{C}\bar{B}u_0, \quad y_1 = \bar{C}\bar{A}\bar{B}u_0 + \bar{C}\bar{B}u_1, \quad ..., \quad y_{L-1} = \sum_{i=0}^{L-1} \bar{C}\bar{A}^{L-1-i}\bar{B}u_i$$

Where $N$ is the state size. Inputs and outputs are scalars.

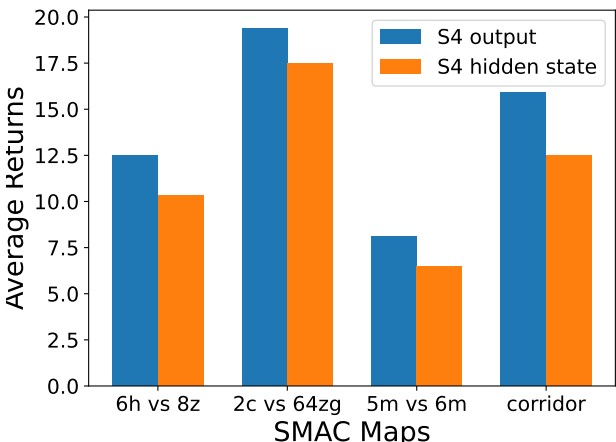

Figure 7: Average Returns obtained in SMAC tasks by passing S4 output versus S4 latent states.

Since the recurrent rule is linear, it can be computed in closed form with matrix multiplication or non-circular convolution:

$$
\begin{bmatrix} y_0 \\ y_1 \\ \vdots \\ y_{L-1} \end{bmatrix} = \begin{bmatrix} \bar{C}\bar{B} & 0 & 0 & 0 & 0 \\ \bar{C}\bar{A}\bar{B} & \bar{C}\bar{B} & 0 & 0 & 0 \\ \bar{C}\bar{A}^2\bar{B} & \bar{C}\bar{A}\bar{B} & \bar{C}\bar{B} & 0 & 0 \\ \vdots & \vdots & \vdots & \vdots & \vdots \\ \bar{C}\bar{A}^{L-2}\bar{B} & \bar{C}\bar{A}^{L-3}\bar{B} & \bar{C}\bar{A}^{L-4}\bar{B} & ... & \bar{C}\bar{B} \end{bmatrix} \begin{bmatrix} u_0 \\ u_1 \\ \vdots \\ u_{L-1} \end{bmatrix} \tag{13}
$$

i.e., $y = \bar{k} * u$ for some kernel $\bar{k}$, which can be calculated by fixing the sequence length $L$ before training. This kernel can be efficiently computed using FFT operations; for example, (Gu et al., 2022) computes the kernel via inverse FFT on the spectrum of $\bar{k}$, which is calculated via Cauchy kernel and the Woodbury Identity. This benefits from the "Normal Plus Low Rank" parameterization of the HIPPO-initialized state transition matrix $A$, and other more efficient parameterizations are proposed in (Gupta et al., 2022).

The SSM, as represented above, operates on scalars or one channel of inputs. To handle vector inputs $\in \mathbb{R}^H$, $H$ copies of the 1-D SSM layer are stacked, one for each input channel, and a linear mixing layer in the after block of the S4 layer mixes the information from different channels to produce outputs $\in \mathbb{R}^H$.

### A.2 Sharing Hidden State Representations

The raw outputs from the S4 layer consist of $y_k = \bar{C}x_k$, where $y_k \in \mathbb{R}^H$ and the latent states $x_k \in \mathbb{R}^{N \times H}$ for $H$ input channels. Since the outputs are linear projections and offer a compact representation of the latent states (or, memory of the agent), this has been used as the message that is transmitted from one agent to the next in the SE-MDP. This offers several advantages: i) results in better team performance; ii) offers scalable cooperation between agents, which eliminates the need for a centralized transformer or a critic, which requires access to information from all agents; one agent needs access to only its immediate neighbor in the sequence; (iii) allows parallel training via convolution.

We also experimented with passing the raw hidden states $x_k \in \mathbb{R}^{N \times H}$ from one agent to another. The hidden states can be complex, depending on the parameterization of the S4 kernel. Therefore, before passing the latent states directly, we first linearly mix the hidden states across the $H$ channels to obtain $x_k \in \mathbb{C}^N$. Then, we linearly project the real and imaginary parts of $x_k$ after concatenation. This mode of information transfer, however, has notable drawbacks: i) it requires computing the S4 hidden state at every timestep, which requires recurrent rollouts of the S4 kernel, and ii) it fails to outperform the method of passing the S4 outputs; possibly due to errors accumulated during recurrent training. A comparison of performance using

S4 output representation versus S4 latent state representation is shown in Figure7, where passing S4 outputs resulted in better performance across all tasks.

It is, however, noted that hidden states at each timestep may be efficiently obtained utilizing the parallel (associate) scan operation as done in (Smith et al., 2022; Lu et al., 2024), but this requires JAX implementation and is currently not supported by PyTorch.

### A.3 Preliminary study using Mamba

We also explored Mamba as an alternative to LTI S4-based models. Mamba allows time-variant parameters to be considered in the SSM equations. Though convolution cannot be applied here since the kernel cannot be computed apriori since the parameters $B, C$ are input-dependent, efficient parallel scan operation allows for parallelizable $O(\log L)$ complexity. However, preliminary analysis utilizing Mamba resulted in suboptimal performance, and it requires more extensive analysis.

### A.4 Experimental Setup and Training

In all experiments, we set the input channel size to $H = 96$ and the S4 state size to $N = 96$. Offline training is conducted on batches of 64 trajectories, with the maximum trajectory length in the offline dataset used as the length for each batch. The shorter trajectories are zero-padded to a constant length. The training was performed using Adam optimizer with a learning rate of $10^{-4}$.

The offline trained model is fine-tuned online using on-policy MAPPO. During the initial stage of fine-tuning, the actor network is kept frozen, and the critic is first trained for the first 50,000 iterations. After this, both the actor and critic are trained simultaneously, with a slower learning rate for the actor network ($10^{-5}$) compared to the critic ($10^{-4}$). During on-policy fine-tuning, the returns-to-go is set at 10% higher than the highest returns encountered during training. On-policy training is conducted in batches of 64. To mitigate the issue of deteriorating performance with prolonged on-policy training, the S4 kernel $A$ can be kept frozen. All experiments were run on a single NVIDIA RTX 2080Ti GPU. Experiments on the RWARE domain take less than 2 hrs to reach optimal performance, and experiments on the SMAC domain take less than 6hrs, 12 hrs, 12 hrs, and 30 hrs for maps 2c vs. 64zg, 5m vs. 6m, 6h vs. 8z and Corridor, respectively.

## B Datasets and Baselines

### B.1 Multi-RobotWarehouse (RWARE)

The offline dataset on RWARE (Papoudakis et al., 2020) is obtained from (Matsunaga et al., 2023), which contains an expert dataset with diverse behaviors obtained by training MAT on small and tiny maps. The dataset consists of 1000 trajectories, each trajectory consisting of 500 timesteps. The dataset statistics are in Table 4. The longest trajectories consist of timesteps in the range of 500 in all the datasets.

The baseline results are obtained from (Matsunaga et al., 2023), which currently holds the state-of-the-art results of the baselines listed on this dataset.

| Map Name | Maximum | Minimum | Average |
|---|---|---|---|
| small 2 agents | 12.37 | 1.13 | 7.12 |
| small 4 agents | 12.08 | 3.93 | 9.49 |
| small 6 agents | 12.69 | 7.59 | 10.76 |
| tiny 2 agents | 16.81 | 1.97 | 12.77 |
| tiny 4 agents | 18.63 | 10.40 | 15.67 |
| tiny 6 agents | 19.97 | 11.88 | 17.45 |

Table 4: RWARE datasets

## B.2 SMAC

The offline SMAC (Samvelyan et al., 2019) dataset is obtained from (Wang et al., 2024). This dataset is obtained by randomly sampling 1000 trajectories from the original dataset provided by (Meng et al., 2021). We consider 4 representative battle maps, including 2 hard maps (5m vs 6m, 2c vs 64zg) and 2 super hard maps (6h vs 8z, corridor), which are detailed in Table 5. The average returns for the dataset are listed in Table 6. The longest trajectories are encountered in the Corridor map, which typically comprises about 100 timesteps.

| Map Name | Ally Units | Enemy Units | Type |
|---|---|---|---|
| 5m_vs_6m | 5 Marines | 6 Marines | homogeneous & asymmetric |
| 2c_vs_64zg | 2 Colossi | 64 Zerglings | micro-trick: positioning |
| 6h_vs_8z | 6 Hydralisks | 8 Zealots | micro-trick: focus fire |
| corridor | 6 Zealots | 24 Zerglings | micro-trick: wall off |

Table 5: SMAC maps for experiments.

| Map Name | Quality | Average Return |
|---|---|---|
| 5m_vs_6m | good | 20.00 |
| | medium | 11.03 |
| | poor | 8.50 |
| 2c_vs_64zg | good | 19.94 |
| | medium | 13.00 |
| | poor | 8.89 |
| 6h_vs_8z | good | 17.84 |
| | medium | 11.96 |
| | poor | 9.12 |
| corridor | good | 19.88 |
| | medium | 13.07 |
| | poor | 4.93 |

Table 6: SMAC datasets.

The offline RL-based baseline results are obtained from (Wang et al., 2024), and MADT results are obtained by running the code available with (Meng et al., 2021).

# C  Additional Analysis

## C.1  Effect of freezing $A$ during on-policy fine-tuning

The degrading effect on MADS4 performance during recurrent on-policy fine-tuning can be mitigated by freezing the S4 kernel parameter $A$ while updating only parameters $B$ and $C$, as illustrated in Figure 8. A similar observation has also been reported in (Bar-David et al., 2023).

## C.2  Effect of global states as inputs

Building on prior work such as MADT, the proposed S4-based MADS4 agents utilize global states as inputs. However, in certain environments, access to the global state may be restricted or unavailable. To address this, we present an ablation study (Figure 9) evaluating the impact of using global state variables as inputs. The results indicate that omitting the global state does not lead to a significant drop in performance.

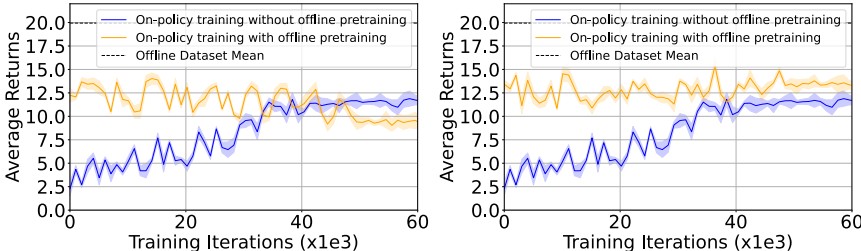

Figure 8: The effect of freezing S4 kernel parameter $A$ in the SMAC 6h vs 8z map. Freezing $A$ in the right Figure results in more stable performance during the on-policy recurrent fine-tuning.

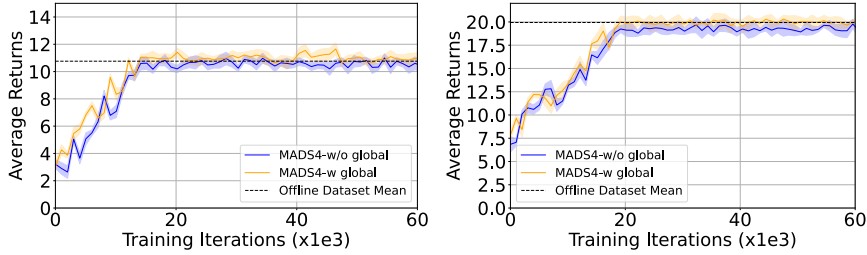

Figure 9: Performance comparison on RWARE small map with 6 agents (left) and SMAC map 2c vs 64 zg (right). The results demonstrate that excluding global states as inputs in MADS4 agents has minimal impact on performance.

