# OpenReview forum: "Multi-Agent Decision S4: Leveraging State Space Models for Offline Multi-Agent Reinforcement Learning"
_TMLR — Rejected by TMLR_

### Review · Reviewer_Etyu · 2025-04-22

**Summary Of Contributions:**

Structured State Space Sequence (S4) models have shown promise for learning
long-term dependencies in single-agent offline reinforcement learning on
sequence-based data. The main contribution of this paper is the novel
adaptation of the S4 model to the multi-agent RL setting.

While the mechanism by which the S4 model can be trained on sequential data in
RL is not original, the authors show how multiple S4-based agents can be trained
to share information in order to take actions cooperatively. The authors show
that the information sharing can also be done in a limited way to reduce the
number of agents that need to share with each other (the information is passed
hierarchically and so each agent only needs information from only one other
agent).

The authors demonstrate that their proposed approach (MADS4) outperforms other existing
algorithms in the offline learning task in two different domains. Furthermore,
they show that MADS4 also improves online fine-tuning of offline training.

**Audience:**

Yes

**Broader Impact Concerns:**

I have no concerns in this regard.

**Claims And Evidence:**

Yes

**Requested Changes:**

None of the requested changes here are critical however I would recommend making
them. The only change strengthening acceptance would be the one requested in the
previous section.

- On page 2, the authors state that value-based regularizations aim to learn
conservative value functions. Explain conservative in this context.

- Page 2 again, the last paragraph reads a little oddly because of the phrase,
"on the other hand". Is DT an alternative regularization technique or an
alternative paradigm?

- Use \textcite{} to refer to authors, not \cite or \parencite.

- Explain returns-to-go. Is it just the expected return? Or how is it different?

- $N$ and $L$ are not defined.

- SSM is not defined.

- Why is Fig 2 in Sec 3.3? It has nothing to do with the explanation but it did
  distract and confuse me.

- Is $s^t_{gi}$ different for each agent? If so, how is it "global"?

- It is unclear what the lines in Fig 1 are. Perhaps add an explanation of them
  and/or make them more distinct rather than just in color.

- "During exploration, the desired returns-to-go is set at 10\% higher than the
  current model's highest return." Justify this choice.

**Strengths And Weaknesses:**

The paper, for the most part, has clear explanations, a strong motivation and an
intuitively reasonable approach for designing MADS4.

The evaluation criteria and tasks chosen to demonstrate the practical performance
of MADS4 is appropriate.

The baselines against which MADS4 is compared are also appropriate. The results
also support the claims made by the authors.

However, while most of the paper is well-written, sections 3.3 and 3.4 were very
difficult to understand. While I am not familiar with the literature in
structured state space modelling, I feel that these sections did not adequately
explain the concept to understand MADS4. I had to quickly go through a few of
the cited references to gain some intuition. I have no specific comments on how
to change these sections but I definitely beleive the explanation needs to be
improved. For instance, if one doesn't know state space modelling $u, x,
\dot{x}, y$ need to be defined. Secondly, a quick explanation of the matrices
$\mathbf{A}, \mathbf{B}, \mathbf{C}, \mathbf{D}$ would be useful.

---

> ### Author Response · Authors · 2025-06-11
> **Response to reviews**
>
> > **Comment** However, while most of the paper is well-written, sections 3.3 and 3.4 were very difficult to understand. While I am not familiar with the literature in structured state space modelling, I feel that these sections did not adequately explain the concept to understand MADS4. I had to quickly go through a few of the cited references to gain some intuition. I have no specific comments on how to change these sections but I definitely beleive the explanation needs to be improved. For instance, if one doesn't know state space modelling $u, x, \dot{x}, y$ need to be defined. Secondly, a quick explanation of the matrices $\mathbf{A}, \mathbf{B}, \mathbf{C}, \mathbf{D}$ would be useful.
>
> **Response**
> Thank you for your valuable feedback. We have improved the readability of Sections 3.3 and 3.4 by adding clearer definitions and background for structured state space models. Additionally, we have expanded the details on S4 modeling in the Appendix.
>
> We have now added a Problem Formulation (Section 3.1) for our multi-agent decision making framework. Therefore, the previous sections 3.3 and 3.4 are now 3.4 and 3.5 respectively. All additional text and/or changes are added in blue font.
>
> > **Comment**
> On page 2, the authors state that value-based regularizations aim to learn conservative value functions. Explain conservative in this context.
>
> **Response**
> In offline reinforcement learning (RL), the agent learns from a fixed dataset of past interactions. However, during training, it may encounter out-of-distribution (OOD) state-action pairs—combinations that are not well represented in the dataset.
>
> Value-based regularization techniques are designed to prevent the agent from overestimating the value of such unfamiliar (OOD) actions or states. They do this by making the learned value function more conservative—that is, assigning lower value estimates to actions that the policy hasn’t seen enough data for. This reduces the risk of the policy exploiting value overestimation and helps maintain stability during offline learning.
>
> > **Comment**
> Page 2 again, the last paragraph reads a little oddly because of the phrase, "on the other hand". Is DT an alternative regularization technique or an alternative paradigm?
>
> **Response**
> Thank you for this comment. DT is an alternative paradigm in offline RL which takes a sequence-learning based approach rather than relying on value or policy-based regularizations. We have rephrased this line in the paper to reflect this better.
>
> > **Comment**
> Explain returns-to-go. Is it just the expected return? Or how is it different?
>
> **Response**
> Thank you for this question. Returns-to-go is related to the concept of expected return, but with a specific conditioning structure used in sequence-based models like Decision Transformer (DT).
>
> This is different from the expected return, which represents the expected cumulative reward over trajectories sampled from a policy. In contrast, RTG is a deterministic value computed from a specific offline trajectory.
>
> The return-to-go (RTG) refers to the cumulative future reward from a given timestep $t$ onward, and is defined as $\sum_{i=t}^{T} r_i$.  In sequence-based offline RL methods such as Decision Transformer, RTG serves as a conditioning signal—the model is trained to predict actions based on the desired return-to-go, past observations, and previous actions. This framing allows the model to learn goal-directed behaviors directly from offline data, without relying on value estimation or policy regularization. We have added a few lines in Section 3.3 to make this clear.
>
> > **Comment**
> $N$ and $L$ are not defined.
>
> **Response**
> Thank you for pointing this out. For better clarity, we have replaced N with T. We have clearly defined T (in Section 3.3) as the end of the horizon of the decision-making problem. L is also defined (in Section 3.4) as the context length in the structured state space (S4) model which is fixed during training.
>
> > **Comment**
> SSM is not defined.
>
> **Response**
> Thank you for rightly pointing this out. We have now defined SSM in Section 3.4.
>
> > **Comment**
> Why is Fig 2 in Sec 3.3? It has nothing to do with the explanation but it did distract and confuse me.
>
> **Response**
> Thank you for this comment. We have moved this figure to Section 4.2, where we discuss about the architecture of the MADS4 agent.
>
> > **Comment**
> Is $s^t_{gi}$ different for each agent? If so, how is it "global"?
>
> **Response**
> Thank you for this question. In typical MARL environments, all agents often have access to the same global state $s^t_g$, which represents the full environment configuration. Therefore, $s^t_{gi}$ will be the same for each agent. However, we use the subscript
> $i$ to allow for a more general formulation where each agent could, in principle, receive a different version of the global state.

---

> > ### Author Response · Authors · 2025-06-11
> > **Continued Response**
> >
> > > **Comment**
> > It is unclear what the lines in Fig 1 are. Perhaps add an explanation of them and/or make them more distinct rather than just in color.
> >
> > **Response**
> > Thank you for this comment. We have added more detail in the caption related to the different colors shown for lines in forward direction and backward (gradient) direction.
> >
> > > **Comment**
> > "During exploration, the desired returns-to-go is set at 10% higher than the current model's highest return." Justify this choice.
> >
> > **Response**
> > Thank you for the question. Setting the desired return-to-go to 10% above the current model's highest return is a deliberate design choice to encourage optimistic exploration and guide the policy toward discovering higher-return trajectories not present in the offline dataset. The choice of 10% also aligns with the strategy used in the Decision S4 paper, where such goal-directed conditioning was shown to improve performance during on-policy fine-tuning. In our case, it helps the model avoid plateauing and promotes continuous policy improvement.

---

### Review · Reviewer_HDsE · 2025-04-27

**Summary Of Contributions:**

The paper proposes an S4-based architecture to train offline a mutli-agent RL algorithm. One of the  key ideas is to order the agent (more or less arbitrarily) and limit the communications on the resulting chain, that is between consecutive agents. This means that the sequence fed to the S4 model for training includes the memory (latent state) of the predecessor agent only. The take-away from the paper is that such S4 architecture is at least as good as, and potentially better than, transformers for this multi-agent task.

**Audience:**

Yes

**Claims And Evidence:**

Yes

**Requested Changes:**

## Questions:

* In Offline MADS4, the parameter update of \theta is within the t loop wihtin the agent i loop. However, the update equation does not clearly depend on the t variable of the loop. It may be a notation issue (typo), but then I am not sure if you update the parameter \theta at each step of the trajectory.
* Figure 3: The online fine-tuning (on top of offline training) seems to have little to no impact on the performance on some tasks. It’s very clear on the 2 middle plots where the yellow line is essentially flat, but I’d argue it’s true on the 4 plots up to the randomness over 5 runs. It’s a bit surprising that no additional performance can be gained from sampling online trajectories, unless the datasets already have a broad and high-quality coverage. I am not sure I am convinced by the comments on this paragraph.
* Figure 4 seems to show that the additional information sharing as proposed (restricted on a chain graph) bring some improvement but it’s quite restricted. Do you think more communication would improve results further?
* You study the effect of the agent order but your conclusions indicate that either you don’t trust your results, or you don’t trust your methodology: “Nonetheless, we recommend using a random order during training to avoid introducing potential biases into the learning process.” I would actually agree with this sentence but your experiment would rather tend to show that this effect doesn’t exist. Could you not exhibit an ordering that actually creates a bias? Maybe a well chosen one. What you want to say is that randomly choosing one specific ordering during training is risky. So to be convinced, I’d need to see that there exists at least one bad one, since on average it seems to not hurt much.
*
## Minor comments:

* Equation 8: a \leq would be nicer than this <=
* Table 3, you name it “Results” but are you reporting loss (the lower the better) or reward?
* I am not exactly from this community and this description of the maps is not understandable for me: “we chose four representative maps consisting of two hard (5m vs. 6m, 2c vs. 64zg) and two super hard (6h vs. 8z, corridor) maps”. What do “5m vs 6m” etc. mean? what do the letters correspond to? “super hard” is a slightly awkward qualification for task complexity, but maybe that’s an accepted terminology in the field? I have a more theory background and I tend to be used to more quantifiable complexities so I was a bit confused, but I decided to read your results more qualitatively, hoping a more expert reviewer could comment more specifically on that. I would still think it’d be better to be more specific on the choice of benchmark and the reasons why it is “hard” or “super hard”.

**Strengths And Weaknesses:**

## Strengths:
* The problem and working hypotheses are important and relevant to the ML community. Namely, the question of studying the adequation to RL tasks of various architectures for sequence predictions.
* The contributions and results are good (table 1 and 2) and empirical results are obtained on large-scale and challenging benchmarks.

## Weaknesses:
* The paper is fairly well written, even though it relies on a decent amount of background on MARL and details on architectures that I am not sure I fully got.  Some parts of the related work and intro could be written a bit more specifically and clearly.
* I have a number of questions below that I would like to discuss to assess the significance and impact of the take-aways from the paper. Some experiments seem to be not fully conclusive.

---

> ### Author Response · Authors · 2025-06-12
> **Response to reviews**
>
> > **Comment**
> The paper is fairly well written, even though it relies on a decent amount of background on MARL and details on architectures that I am not sure I fully got. Some parts of the related work and intro could be written a bit more specifically and clearly.
>
> **Response**
> Thank you for this comment. We have improved the Methodology section to show the details of the architectures and models used in our work. We are also in the process of rewriting parts of the Intro and Related Work more clearly.
>
> > **Comment**
> In Offline MADS4, the parameter update of \theta is within the t loop within the agent i loop. However, the update equation does not clearly depend on the t variable of the loop. It may be a notation issue (typo), but then I am not sure if you update the parameter \theta at each step of the trajectory.
>
> **Response**
> Thank you for rightly pointing this out. We have now revised the algorithm to clarify that $\theta$ is updated once per trajectory, not at each timestep. Specifically, for each agent, the predicted action sequence is compared against the target action sequence from the offline dataset, and the loss is computed over the entire trajectory. This trajectory-level loss is then used to update the agent’s parameters. This process is repeated for each agent.
>
> > **Comment**
> Figure 3: The online fine-tuning (on top of offline training) seems to have little to no impact on the performance on some tasks. It’s very clear on the 2 middle plots where the yellow line is essentially flat, but I’d argue it’s true on the 4 plots up to the randomness over 5 runs. It’s a bit surprising that no additional performance can be gained from sampling online trajectories, unless the datasets already have a broad and high-quality coverage. I am not sure I am convinced by the comments on this paragraph.
>
> **Response**
> Thank you for this insightful comment. We agree that the benefit of on-policy fine-tuning in our setting is modest, and there are several factors contributing to this. First, as also noted in the review, the offline datasets used for pretraining consist of high-quality (expert) trajectories. As a result, the pretrained models already achieve strong performance, leaving limited room for improvement during fine-tuning. Second, while the online phase is trained using MAPPO—which promotes exploration through entropy regularization—this additional exploration has limited effect since the models are already operating near-optimal behaviors learned from offline data. Third, the online phase of MADS4 leverages the recurrent formulation of S4-based models, which, as previously observed in related work (e.g., Decision S4), can be sensitive and prone to instability. To ensure stable performance during on-policy fine-tuning, we freeze the S4 transition matrix A\mathbf{A}A, which mitigates divergence and performance degradation. While this reduces the number of learnable parameters and limits the potential for further improvement, it results in more stable and consistent fine-tuning performance, as shown in Appendix C.1.
>
> We would also like to clarify that while the improvements may appear modest in absolute terms, they are meaningful and significant within the context of these benchmark environments, where state-of-the-art algorithms often differ by only small margins.
>
> > **Comment**
> Figure 4 seems to show that the additional information sharing as proposed (restricted on a chain graph) bring some improvement but it’s quite restricted. Do you think more communication would improve results further?
>
> **Response**
> Thank you for this question. While the performance improvements reported may appear marginal in absolute terms, they are in fact significant given the maturity and difficulty of the benchmark tasks—where even small gains are meaningful.
>
> In our work, we adopt the Sequentially-Expanded MDP (SE-MDP) formulation, where each agent shares information only with its immediate predecessor. We also experimented with a setting where each agent receives information from all preceding agents in the sequence. However, this provided no additional benefit. This outcome is expected, as the memory state of an agent already encodes information propagated from all previous agents; explicitly passing information from each prior agent introduces redundancy without contributing new signal.
>
> While there are existing approaches that learn optimized communication graphs to improve coordination, they typically involve higher complexity and do not scale well to large numbers of agents. In contrast, our simple and efficient pairwise information flow enables scalable and effective cooperation, achieving strong performance without requiring complex communication structures.

---

> > ### Author Response · Authors · 2025-06-12
> > **Continued Response**
> >
> > > **Comment**
> > You study the effect of the agent order but your conclusions indicate that either you don’t trust your results, or you don’t trust your methodology: “Nonetheless, we recommend using a random order during training to avoid introducing potential biases into the learning process.” I would actually agree with this sentence but your experiment would rather tend to show that this effect doesn’t exist. Could you not exhibit an ordering that actually creates a bias? Maybe a well chosen one. What you want to say is that randomly choosing one specific ordering during training is risky. So to be convinced, I’d need to see that there exists at least one bad one, since on average it seems to not hurt much.
> >
> > **Response**
> > Thank you for the thoughtful observation. You are correct that our experimental results do not show a significant drop in performance across different agent orderings. We evaluated both fixed and random orderings in the SE-MDP setting and did not identify a specific ordering that consistently degraded performance.
> >
> > There are two main reasons for this outcome. First, our MADS4 framework employs a centralized training approach where each agent, despite only directly receiving the hidden state from its immediate predecessor, indirectly benefits from information propagated throughout the sequence. Since the hidden state of each agent encodes context from previous agents, even fixed-order training can allow meaningful signal flow between all agents.
> >
> > Second, the benchmark environments we use (e.g., RWARE, SMAC) emphasize general cooperation rather than strict inter-agent dependencies or role-based hierarchies. That is, success typically depends on aggregate coordination rather than specific ordering or sequential dependencies among agents. This observation is also supported by prior work such as the ACE framework (Li et al., 2023), which similarly found that ordering had limited impact in SE-MDP-style training.
> >
> > That said, we agree that relying on a single fixed order may be risky in environments with stronger asymmetries or hierarchical coordination requirements. Our recommendation to randomize the order during training is thus a precautionary strategy—not because our experiments suggest a strong ordering effect, but because randomization provides robustness and prevents potential bias from any untested or domain-specific ordering that could introduce implicit dependencies.
> >
> > > **Comment**
> > Equation 8: a \leq would be nicer than this <=
> >
> > **Response**
> > Thank you for catching this, we fixed it.
> >
> > > **Comment**
> > Table 3, you name it “Results” but are you reporting loss (the lower the better) or reward?
> >
> > **Response**
> > Thank you for pointing this out. Results show the mean average returns (the higher the better) for different model sizes and we also show the % of parameters used in MADS4 as compared to the number of parameters used in a competent transformer-based model (MADTKD). We further clarify this in the Table 3 caption.
> >
> > > **Comment**
> > I am not exactly from this community and this description of the maps is not understandable for me: “we chose four representative maps consisting of two hard (5m vs. 6m, 2c vs. 64zg) and two super hard (6h vs. 8z, corridor) maps”. What do “5m vs 6m” etc. mean? what do the letters correspond to? “super hard” is a slightly awkward qualification for task complexity, but maybe that’s an accepted terminology in the field? I have a more theory background and I tend to be used to more quantifiable complexities so I was a bit confused, but I decided to read your results more qualitatively, hoping a more expert reviewer could comment more specifically on that. I would still think it’d be better to be more specific on the choice of benchmark and the reasons why it is “hard” or “super hard”.
> >
> > ** Response**
> > Thank you for raising this important point. We acknowledge that the naming conventions and difficulty levels of SMAC (StarCraft Multi-Agent Challenge) maps can be unclear to those outside the MARL community.
> > The map names follow a shorthand convention indicating the unit types and team sizes involved. For example:
> >
> > •	"5m vs. 6m" means 5 Marines (our agents) versus 6 enemy Marines.
> >
> > •	"2c vs. 64zg" means 2 Colossi (agents) versus 64 Zerglings (enemies).
> >
> > •	"6h vs. 8z" involves 6 Hydralisks versus 8 Zealots.
> >
> > •	"corridor" is a named map that places agents in a narrow space, making coordinated movement and targeting significantly more challenging.
> >
> > In the SMAC benchmark, tasks are empirically categorized as easy, hard, and super hard based on their observed success rates and learning difficulty across a wide range of algorithms. These labels are widely used in the community to indicate the relative complexity of coordination and planning required.

---

> ### Comment · Reviewer_HDsE · 2025-06-29
> **Thank you for your response**
>
> Your responses are clear and the updates to the draft are clarifying most of my doubts. Again, as a non-direct expert in this field, I can mostly assess the validity of the methodology and consistence of the conclusions, and now I am convinced by the findings.
>
> I have a small questions: will you make the code available?

---

> > ### Author Response · Authors · 2025-06-29
> >
> > We greatly appreciate your feedback and are glad that the revisions have clarified your doubts and strengthened the presentation of our work.
> >
> > Regarding your question on code availability: yes, we plan to make the code publicly available upon publication of the paper. We will include the link to the repository in the final version to ensure reproducibility.

---

> > > ### Comment · Reviewer_HDsE · 2025-07-07
> > > **significance of the results and adequation of the claims**
> > >
> > > The discussions with the reviewers and AC makes me realise that one of the weaknesses I highlight may have not been fully addressed by your rebuttal. Namely I state that "I would like to discuss to assess the *significance and impact* of the take-aways from the paper. Some *experiments seem to be not fully conclusive*."
> > >
> > > This issue has also been raised by other reviewers (not enough seeds to conclude). It appears that your claims in your abstract and discussions of your results sound too assertive. For instance, in your abstract, you say "Experiments [...] demonstrate that our approach significantly *outperforms state-of-the-art offline RL* and transformer-based MARL baselines across most tasks." But I think your methodology may not quite show something so strong.
> > >
> > > Can you please comment on this? Right now I am not sure your results are really statistically significant.

---

> > > > ### Author Response · Authors · 2025-07-12
> > > >
> > > > Thank you for your comment on this. We evaluated the performance of MADS4 on our benchmark environments by training the algorithm with 5 random seeds and reporting the mean and standard deviation of average returns obtained over 30 independent evaluation episodes. This evaluation protocol is consistent with the baselines used in this work, such as OMIGA [1], OMAR [2], and MADT [3].
> > > >
> > > > In order to further assess the performance of MADS4, we increased the number of training seeds to 10 and evaluated its performance on the Warehouse domain. The results are summarized below.
> > > >
> > > > ### Table: Average returns and standard deviations over 10 random seeds on the Warehouse domain
> > > >
> > > > | Method         | Tiny (11x11) (N=2) | Tiny (11x11) (N=4) | Tiny (11x11) (N=6) | Small (11x20) (N=2) | Small (11x20) (N=4) | Small (11x20) (N=6) |
> > > > |----------------|--------------------|--------------------|--------------------|---------------------|---------------------|---------------------|
> > > > | BC             | 8.80 ± 0.25        | 11.12 ± 0.19       | 14.06 ± 0.32       | 5.54 ± 0.06         | 7.88 ± 0.14         | 8.90 ± 0.13         |
> > > > | ICQ            | 9.38 ± 0.75        | 12.13 ± 0.44       | 14.59 ± 0.16       | 5.43 ± 0.19         | 7.93 ± 0.19         | 8.87 ± 0.22         |
> > > > | OMAR           | 6.77 ± 0.64        | 14.39 ± 0.91       | 16.13 ± 1.21       | 4.40 ± 0.34         | 7.12 ± 0.38         | 8.41 ± 0.49         |
> > > > | MADTKD         | 6.24 ± 0.60        | 9.90 ± 0.21        | 13.06 ± 0.19       | 3.65 ± 0.34         | 6.85 ± 0.36         | 7.85 ± 0.52         |
> > > > | OptiDICE       | 8.70 ± 0.06        | 11.13 ± 0.44       | 14.02 ± 0.36       | 4.84 ± 0.32         | 7.68 ± 0.09         | 8.47 ± 0.26         |
> > > > | AlberDICE      | **11.15 ± 0.35**   | 13.11 ± 0.32       | 15.72 ± 0.36       | 5.97 ± 0.11         | 8.18 ± 0.19         | 9.65 ± 0.13         |
> > > > | **MADS4 (ours)** | **11.81 ± 0.86** | **15.31 ± 0.41**   | **17.63 ± 0.92**   | **6.48 ± 0.34**     | **9.77 ± 0.35**     | **10.69 ± 0.78**    |
> > > >
> > > > The MADS4 shows improved performance across almost all configurations of the Warehouse domain. In the SMAC domain, MADS4 demonstrates competitive performance; however, since it does not outperform baselines on all SMAC maps, we have revised our claims in the abstract to reflect this.
> > > >
> > > > “Experiments on challenging MARL benchmarks, including Multi-Robot Warehouse (RWARE) and StarCraft Multi-Agent Challenge (SMAC), demonstrate that our approach achieves competitive or improved performance compared to state-of-the-art offline RL and transformer-based MARL baselines across most tasks.”
> > > >
> > > > [1] Wang, Xiangsen, et al. "Offline multi-agent reinforcement learning with implicit global-to-local value regularization." Advances in Neural Information Processing Systems 36 (2023): 52413-52429.
> > > >
> > > > [2] Pan, Ling, et al. "Plan better amid conservatism: Offline multi-agent reinforcement learning with actor rectification." International conference on machine learning. PMLR, 2022.
> > > >
> > > > [3] Meng, Linghui, et al. "Offline pre-trained multi-agent decision transformer." Machine Intelligence Research 20.2 (2023): 233-248.

---

> ### Comment · Reviewer_7hDV · 2025-07-17
>
> More seeds is a step in the right direction, but I'd like to see even more seeds and a meaningful statistical comparison - e.g. seeing if there are (not) overlapping confidence intervals between MADS4 and other approaches.

---

> ### Author Response · Authors · 2025-07-17
>
> Thank you for highlighting this. All baseline results used for comparison with MADS4 were sourced from their respective papers, each of which reported performance across 5 training seeds (due to the training cost associated with training each method). To ensure a fair comparison, we adhered to the same evaluation protocol. In response to your previous comment, we used a higher number of training seeds, and did not see much difference in the results (confidence intervals).
>
> To aid interpretation, we highlight in bold those methods whose performance confidence intervals overlap with that of MADS4. In the SMAC domain, MADS4 demonstrates competitive performance relative to existing baselines across all maps. Notably, in the Warehouse domain—which demands greater cooperation, particularly as the number of agents increases—MADS4 consistently outperforms other methods across most scenarios.

---

### Review · Reviewer_7hDV · 2025-05-19

**Summary Of Contributions:**

This paper introduces MADS4: an offline RL algorithm which builds upon the S4 architecture. The main novelty is this architecture allows agents to pass their hidden state to the next agent in a fixed ordering, which allows for additional communication beyond actions. This approach is evaluated on RWARE and SMAC benchmarks, with improved performance over other algorithms. The authors also conduct ablation studies to see the effect of the ordering of agents and the effect of sharing information via the latent representation.

**Audience:**

Yes

**Claims And Evidence:**

Yes

**Requested Changes:**

## Critical for Recommendation

**Hidden State**. I would really like to see more experiments and analysis around the sharing of hidden states between agents, which seems like the primary contribution of this paper.
*  In practice, some applications of MARL prevent agents from sharing information without additional infrastructure, so I could see this approach not being totally general. I would like to see a discussion around when it is reasonable to have this sort of communication protocol and cases where it is not available.
* If the communication mechanism is not available during deployment, can we still use MADS4 to train agents? I'd like to see the experiments about the removal of the ability to share hidden state between offline training and online deployment and how this impacts performance.


**Ordering of Agents**
The authors conclude that the ordering of agents has no effect on performance. I would like to see some discussion or experimental analysis (even on a toy environment) about whether this claim holds generally. I could imagine in some environments ordering does matter. For example, if agents 1 and 3 need to tightly coordinate, but neither needs to coordinate with agent 2; an ordering 1-> 2-> 3 is likely worse than 1-> 3-> 2 or 2-> 1-> 3 since 1 and 3 cannot directly communicate and instead must do so via 2.


## Minor
* It would be interesting to see what is actually being communicated in the hidden state. Are there situations where we can see agents communicating?
* Can you show experimentally that agents earlier in the ordering learn to pass on the necessary information needed by later agents?

**Strengths And Weaknesses:**

## Strengths:
* The paper is well-written.
* I think the decomposition of communication into a linear flow of information is an interesting idea that would extend to other multi-agent learning algorithms

## Weaknesses:
* I don't think the authors treat their novel contributions to enough analysis or discussion. For example, this work proposes sharing of information via hidden state but there is no discussion of why this is reasonable. Sharing the hidden state during offline training is possible due to centralized training, but what happens if there is no mechanism for such communication online?
* Experiments are conducted with a relatively low number of seeds.

---

> ### Author Response · Authors · 2025-06-13
> **Response to Reviews**
>
> > **Comment**
> In practice, some applications of MARL prevent agents from sharing information without additional infrastructure, so I could see this approach not being totally general. I would like to see a discussion around when it is reasonable to have this sort of communication protocol and cases where it is not available.
>
> **Response**
> Thank you for this comment. The MADS4 framework falls under the category of MARL with limited communication, similar to approaches such as ACE (Cooperative Multi-Agent Q-learning with Bidirectional Action-Dependency) and CommNet (Learning Multiagent Communication with Backpropagation). While these methods allow for global or full-message communication between every agent in the system, MADS4 is designed to be more lightweight and scalable: each agent communicates only with its immediate neighbor in the sequence. This significantly reduces communication overhead, as only point-to-point information passing is required rather than a full broadcast.
>
> In scenarios where no direct communication channel is available during execution, a potential extension would be to train each agent to predict or distill the latent information of its predecessor based on its local observations. In this way, each agent can still operate under the design conditions of MADS4 with its own prediction of the latent information of the other agent. However, as with any method that relies on inter-agent communication during training or execution, there may be a performance gap when deployed in settings with no communication.
>
> > **Comment**
> If the communication mechanism is not available during deployment, can we still use MADS4 to train agents? I'd like to see the experiments about the removal of the ability to share hidden state between offline training and online deployment and how this impacts performance.
>
> **Response**
> We believe this comment is somewhat related to the previous comment. MADS4 is specifically designed to operate under limited communication, where each agent shares information only with its immediate neighbor in the sequence. However, it does assume the presence of this minimal communication channel during both training and deployment.
>
> If the agent-to-agent communication is available during training but removed at deployment, we would expect a drop in performance, as is the case with any communication-based MARL method. This is because the model relies on the information passed through latent states to coordinate behavior. Having said that, it may be possible to explore communication-distillation strategies—such as training agents to infer the latent states of others—we consider this an interesting direction for future work.
>
> > **Comment**
> The authors conclude that the ordering of agents has no effect on performance. I would like to see some discussion or experimental analysis (even on a toy environment) about whether this claim holds generally. I could imagine in some environments ordering does matter. For example, if agents 1 and 3 need to tightly coordinate, but neither needs to coordinate with agent 2; an ordering 1-> 2-> 3 is likely worse than 1-> 3-> 2 or 2-> 1-> 3 since 1 and 3 cannot directly communicate and instead must do so via 2.
>
> **Response**
>  Thank you for this insightful comment. In our experiments, we did not identify any specific agent ordering that led to significantly worse performance. One key reason for this is the design of MADS4 and the properties of the S4 model.
>
> Although each agent in MADS4 receives hidden state information only from its immediate predecessor, the centralized training setup ensures that the information from earlier agents is propagated through the sequence. S4 models are designed to capture long-range dependencies in sequences, which means that the hidden state passed from agent 2 to agent 3 can still carry relevant information originating from agent 1.
>
> This is analogous to how S4 handles long sequences in natural language or time series modeling—later outputs can attend to and be influenced by earlier inputs through the structured memory dynamics of the model. As a result, even in a sequential agent ordering such as 1 → 2 → 3, the model can effectively incorporate information from agent 1 when computing actions for agent 3.

---

> > ### Author Response · Authors · 2025-06-13
> > **Continued Response**
> >
> > > **Comment**
> > It would be interesting to see what is actually being communicated in the hidden state. Are there situations where we can see agents communicating?
> >
> > **Response**
> > Thank you for this question. In MADS4, each agent is modeled using an S4-based architecture, which maintains a latent memory state that evolves over time and carries information across the sequence. This hidden state acts as the medium for communication between agents in the SE-MDP formulation.
> >
> > For example, in an ordering such as 1 → 2 → 3, agent 1's hidden state encodes information from its own observation-action-return history. Agent 2 then receives this hidden state and incorporates it into its own computation, effectively conditioning its behavior on both its own trajectory and agent 1's summarized information. Similarly, agent 3 receives agent 2’s hidden state, which now contains information aggregated from both agent 1 and agent 2.
> >
> > Thus, while the communication is not explicit or interpretable in the traditional sense, the latent memory state carries task-relevant, compressed representations of upstream agent behavior, enabling implicit communication through the learned dynamics of the S4 model. We have experimented with different representations of the hidden state as the information being passed from one agent to the next in MADS4, shown in Appendix A.2.
> >
> > > **Comment**
> > Can you show experimentally that agents earlier in the ordering learn to pass on the necessary information needed by later agents?
> >
> > **Response**
> > Thank you for the question. The architecture of MADS4 is designed to support this behavior. Specifically, each agent's hidden state in the sequence encodes a compact summary of its own information along with information propagated from all previous agents via the S4 model’s structured memory of its preceding agent in the sequence.
> >
> > During training, later agents are optimized to condition their own action and memory updates on this incoming hidden state. As a result, the learning process encourages earlier agents to encode task-relevant information that benefits downstream agents. This form of implicit communication is enabled by the sequential structure of SE-MDP framework of MADS4 and the long-range dependency modeling capabilities of S4.

---

> ### Comment · Reviewer_7hDV · 2025-07-17
>
> I understand that it is *possible* to pass on information- I'm more curious whether or not this happens empirically. Have you studied the learned representation? Even just looking at the norm of the information passed could give us some sense of whether the agents are truly communicating with the proposed mechanism.

---

> > ### Author Response · Authors · 2025-07-17
> >
> > Thank you for your comment. To evaluate the impact of inter-agent information sharing within the SE-MDP framework of MADS4, we conducted an ablation study using the MADS4 architecture where agents were trained without any information exchange. As shown in Figure 4, removing inter-agent communication leads to a noticeable drop in performance, highlighting the importance of information sharing.
> >
> > We further investigated the type of information being shared by comparing two alternatives (shown in the Appendix): (i) the output of the S4 module, which serves as a linear embedding of the agent’s current action, and (ii) the hidden state of the S4 module, which captures the agent’s internal memory or latent state. As illustrated in Figure 7, sharing the S4 outputs leads to superior performance, as evidenced by consistently higher returns. This can be attributed to the recurrent training needed to obtain the S4 hidden states at each time step, as discussed in Section A.2. We can move this discussion to the main paper.

---

### Decision · Action_Editor_bUFF · 2025-07-23

**Recommendation:** Reject

**Audience:**

Yes

**Audience Explanation:**

Relevant to the MARL community.

**Claims And Evidence:**

No

**Claims Explanation:**

After extensive discussion with reviewers the decision is rejection.

There are issues with the paper that call into question if it meets TMLRs "claims match evidence criteria". In particular, the majority of the results are based on 5 independent runs with standard deviation errorbars (but treated as confidence intervals). Reviewer 7hDV noted this in their review, but the authors did not respond. Reviewer HDsE noted "significance and impact of the take-aways from the paper. Some experiments seem to be not fully conclusive." The authors did not respond, so 7hDV more directly said they did not think the results support the claims of abstract/intro. The authors responded they used 5 seeds because prior work did as well. This is not a valid excuse. if prior published work is flawed/limited, that is not justification for continuing poor empirical practices. The authors did provide a new table of results with 10 seeds, but no explanation for why 10 seeds is enough and more concerningly the errorbars are still standard deviations, which is not a measure of statistical confidence.

It is noted that the claims in the abstract were adjusted to remove the word "significantly".

Reviewer 7hDV then asked about more seeds and confidence intervals, to which the authors responded: "To ensure a fair comparison, we adhered to the same evaluation protocol. In response to your previous comment, we used a higher number of training seeds, and did not see much difference in the results (confidence intervals)." This is not about fairness, this is a matter of correctness. In addition the authors continues to conflate standard deviation with confidence intervals.

Why is this important? The authors are attempting to use empirical evidence as support of the primary contributions of the paper, as stated in the abstract: "Experiments on challenging MARL benchmarks [...] demonstrate that our approach achieves competitive or improved performance compared to state-of-the-art offline RL and transformer-based MARL baselines across most tasks." Without appropriate statistical tools "competitive or improved" is difficult to establish. In fact recent work has demonstrated that ranking claims are extremely difficult to establish [https://dl.acm.org/doi/10.5555/3692070.3692976]. Much has been published lately about how following standard practice is actually quite poor and often misleading.

The results upon submission were far below the bar of claims matching evidence. After the author discussion period, improvements have been made but still short of the bar. Most importantly, the reviewers are not expected to explicitly instruct the authors on exactly what to do to achieve reasonably sound empirical results. I believe this would require several more cycles of comment and response to reach an acceptable level and that is beyond reasonable expectations on the TMLR review process.

This could all be fixed as reviewer 7hDV noted in the discussion:
"Ideally, I'd want to see:
- A meaningful statistical comparison with other benchmarks
- Or targeted experiments showing the utility of the approach. Even on toy environments would suffice."

I suggest the authors do this for the next submission of this work.

**Resubmission Of Major Revision:**

The authors may consider submitting a major revision at a later time.